# Decentralized Control of Quadrotor Swarms with End-to-end Deep Reinforcement Learning

**Sumeet Batra**[*]   **Zhehui Huang**[*]   **Aleksei Petrenko**[*]
**Tushar Kumar**   **Artem Molchanov**   **Gaurav S. Sukhatme**[†]
Department of Computer Science
University of Southern California
{ssbatra,zhehuihu,petrenko,kumart,molchano,gaurav}@usc.edu

**Abstract:** We demonstrate the possibility of learning drone swarm controllers that are zero-shot transferable to real quadrotors via large-scale multi-agent end-to-end reinforcement learning. We train policies parameterized by neural networks that are capable of controlling individual drones in a swarm in a fully decentralized manner. Our policies, trained in simulated environments with realistic quadrotor physics, demonstrate advanced flocking behaviors, perform aggressive maneuvers in tight formations while avoiding collisions with each other, break and re-establish formations to avoid collisions with moving obstacles, and efficiently coordinate in pursuit-evasion tasks. We analyze, in simulation, how different model architectures and parameters of the training regime influence the final performance of neural swarms. We demonstrate the successful deployment of the model learned in simulation to highly resource-constrained physical quadrotors performing station keeping and goal swapping behaviors. Video demonstrations and source code are available at the project website https://sites.google.com/view/swarm-rl.

**Keywords:** Swarms, Multi-robot systems, Multi-robot learning

## 1 Introduction

Teams of unmanned aerial vehicles are widely applicable to *e.g.* area coverage, reconnaissance *etc*. State of the art approaches for planning and control of drone teams require full state information and extensive computational resources to plan in advance [1]. This greatly limits their applicability since existing planning algorithms are often brittle in partially-observable environments and challenging to execute in real time on embedded hardware. Many classical methods also suffer from the curse of dimensionality as the number of possible configurations grows combinatorially with team size. In addition, kinodynamic planners typically require a precise model of drone dynamics.

Here, we present a fully learned approach that uses a small amount of computation during execution and relies exclusively on local observations, yet results in effective controllers for large-scale, swarm-like, teams that are zero-shot transferable to real quadrotors. We take advantage of multi-agent deep reinforcement learning (DRL) to train quadrotor drones on hundreds of millions of environment transitions in realistic simulated environments. We find neural network architectures and observation spaces that allow our neural controllers to achieve high performance on a diverse set of tasks. Our experiments demonstrate that drone swarms controlled by our neural network policies generate highly effective, smooth, and virtually collision-free trajectories. We show simulated dynamic obstacle avoidance and demonstrate that our learned controllers successfully execute tasks on physical drones in team sizes upto 8 quadrotors. To the best of our knowledge, this is the first approach that a) demonstrates scalable coordinated flight of drone swarms in a realistic physical simulation achieved through end-to-end reinforcement learning (RL) with direct thrust control, and b) demonstrates that the learned policies transfer to physical drone teams. We view this as progress towards hardware agnostic real-world deployment of quadrotor swarms with realtime replanning capabilities.

---

[*]Equal contribution

[†]Gaurav Sukhatme holds concurrent appointments as a Professor at USC and as an Amazon Scholar. This paper describes work performed at USC and is not associated with Amazon.

5th Conference on Robot Learning (CoRL 2021), London, UK.

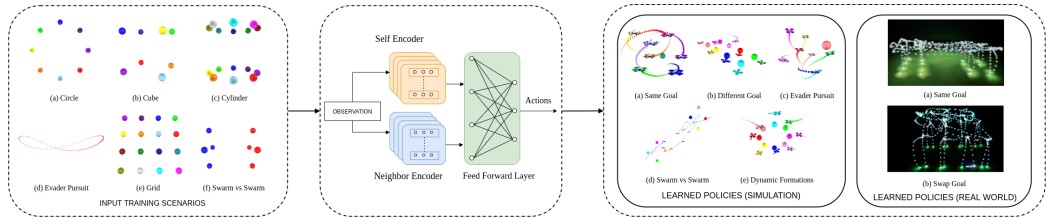

Figure 1: Drones are trained on formations randomly sampled from one of six pre-defined scenarios on the left. Drones employ a self- and neighbor encoder that learn a mapping from a combination of proprioceptive and extereoceptive inputs to thrust outputs. The combined intermediate embeddings enable policies to perform high speed, aggressive maneuvers and learn collision avoidance behaviors *e.g.* formation creation, formation swaps, evader pursuit. These are shown on the right both in simulation and physical experiments.

## 2 Related work

Classical methods such as RRT [2] and PRM [3] can generate collision-free piece-wise linear trajectories. These do not directly account for robot dynamics and scale poorly with team size, limiting their applicability for online re-planning. To account for the dynamics, a smooth trajectory refinement [1] or optimization [4] is performed. Kinodynamic planning methods take advantage of the known dynamics of individual robots and are more suitable for agile and aggressive flight [5]. It is also possible to convert a geometric route to a dynamically feasible trajectory [6] resulting in an agile system or to directly find dynamically feasible paths and refine them with gradient based optimization [7]. Prior learning-based approaches include [8] which estimates reachable states given the quadrotor dynamics. However, these methods require knowledge of the entire state space a priori and do not generalize well to other high dimensional environments. Additionally, kinodynamic planning is typically intractable in the multi-robot setting. Collision avoidance is particularly difficult where drones perform aggressive maneuvers. Traditional techniques use observations from neighboring drones to constrain the geometric free space or the action space of each drone. Prior work includes an $n$-body collision avoidance approach [9] that constrains the velocity space of a robot using velocity information from neighboring robots. Another approach utilizes pose information from neighbors to construct a Voronoi Cell, and each robot plans a collision free trajectory through its respective cell [10]. The approach in [11] minimally constrains the control space of an agent by constructing chance-constrained safety sets that account for measurement noise and disturbances present in real world environments. While these approaches guarantee collision free trajectories and are capable of running in realtime, with the exception of [11] (PrSBC), they significantly constrain the configuration space of the robot, preventing aggressive maneuvers and resulting in occasional deadlock. While PrSBC minimally constrains the control space of any given policy, it requires a model of neighboring agents in order to construct chance-constrained safety sets. Our controllers rely on a reward formulation with a high emphasis on collision avoidance and our policies are not constrained, allowing for aggressive maneuvers in a variety of rich, dynamic environments.

Simulation trained neural networks have been used for single-agent and multi-agent quadrotor control to address shortcomings of kinodynamic planners. [12] and [13] both utilize Imitation Learning (IL) from a global centralized planner, [13] further utilizes RL to balance the tradeoff between actions optimal to the team and individuals. [14, 15] learn controllers robust to aerodynamic interactions. Graph Neural Networks (GNNs) are promising architectures for swarm control and they have been proposed as a parameterization for imitation learning [16, 17] and RL [18, 19] algorithms.

Reinforcement learning has shown promise in learning policies for UAV flight [20]. Deep RL [21] has been used to learn minimum-time trajectory generation for quadrotors. Similar to [20] (but in a multi-agent setting) we train from scratch using DRL via an end-to-end approach: the policies directly control the motor thrust. This is similar to [22] which uses Soft Actor Critic to teach a single quadrotor how to fly. Multi-Agent Reinforcement Learning (MARL) has recently been applied to autonomous driving, path finding, and cooperative multi-agent control [23, 13, 24] and UAV team control (*e.g.* [25] – a new gym environment that models drone dynamics, taking into account aerodynamic effects, and trains policies for basic hovering and leader-following tasks). Khan et al. [26] train a centralized Q-network to solve multi-agent motion planning problems. This approach has limited scalability as the input space of the Q-network increases with the number of drones. We address scaling by relying only on local neighborhood information both during training and inference.

# 3 Method

## 3.1 Problem formulation

We consider a team of $N$ quadrotor drones in a simulated 3D environment. Our task is to learn a control policy that directly maps proprioceptive observations of an individual drone to motor thrusts with the goal of minimizing the distances to positions in the desired formation while avoiding collisions. We analyze the case of online decentralized control, *i.e.* instead of a centralized system solving the joint trajectory optimization problem offline, we consider policies which simultaneously (and implicitly) plan trajectories and control individual quadrotors in real-time. The decentralized approach assumes no access to the global state during evaluation and scales better with the size of the swarm. In the real world, this is analogous to individual quadrotors in the swarm generating their own action sequences using only on-board computations. We demonstrate the efficacy of this approach by showing that our learned policies do indeed transfer to a physical setting. Formally, the state of the environment at time $t$ is described by the tuple $S^{(t)} = (g_1^{(t)}, ..., g_N^{(t)}, s_1^{(t)}, ..., s_N^{(t)})$. Here $g_i^{(t)} \in \mathbb{R}^3$ are the goal locations for the quadrotors that together define the desired swarm formation at time $t$, and $s_i^{(t)}$ are the states of individual quadrotors. We describe the state of a single quadrotor by the tuple $(p, v, R, \omega)$, where $p \in \mathbb{R}^3$ is the position of the quadrotor relative to the goal location, $v \in \mathbb{R}^3$ is the linear velocity in the world frame, $\omega \in \mathbb{R}^3$ is the angular velocity in the body frame, and $R \in SO(3)$ is the rotation matrix from the quadrotor's body coordinate frame to the world frame. Each quadrotor is controlled by a learned policy $\pi_\theta(a^{(t)}|o^{(t)})$ which maps observations $o^{(t)}$ to Gaussian distributions over continuous actions $a^{(t)}$. To effectively maneuver and avoid collisions, the quadrotors need to be able to measure their own position and orientation in space $s_i^{(t)}$, and relative positions $\tilde{p}_{ij}^{(t)}$ and relative velocities $\tilde{v}_{ij}^{(t)}$ of their neighbors. We therefore represent each quadrotor's observations as $o_i^{(t)} = (s_i^{(t)}, \eta_i^{(t)})$, where $\eta_i^{(t)} = (\tilde{p}_{i1}^{(t)}, ..., \tilde{p}_{iK}^{(t)}, \tilde{v}_{i1}^{(t)}, ..., \tilde{v}_{iK}^{(t)})$ is a tuple containing the neighborhood information. Here $K \leq N - 1$ is the number of neighbors that each individual quadrotor can observe. We use $K \ll N$ for larger swarms to improve scalability during both training and evaluation. Simulated quadrotors, similar to their real-world counterparts, are powered by motors that spin in one direction and generate only non-negative thrust. We thus transform actions $a^{(t)} \in \mathbb{R}^4$ sampled by the policy to control inputs $f^{(t)} \in [0, 1]^4$ via $f^{(t)} = \frac{1}{2}(\text{clip}(a^{(t)}, -1, 1) + 1)$, where $f_{1...4}^{(t)} = 0$ corresponds to no thrust, and $f_{1...4}^{(t)} = 1$ corresponds to maximum thrust on motors 1...4.

## 3.2 Simulation and sim-to-real considerations

We train and evaluate our control policies in simulated environments with realistic quadrotor dynamics. We adopt a simulation engine with drone dynamics [20], and augment it to support quadrotor swarms. Virtual quadrotors are modeled on the Crazyflie 2.0 [27] – our physical demonstration platform. A key feature of this engine is the model of hardware imperfections previously shown to facilitate zero-shot sim-to-real transfer of stabilizing policies for single quadrotors [20]. Non-ideal motors are modeled using motor-lag and thrust noise and the simulator provides imperfect observations, with noise injected into position, orientation, and velocity estimations. Noise parameters are estimated from data collected on real quadrotors. While motor and sensor noise create a challenging learning environment, they are instrumental to prevent overfitting to otherwise unrealistic idealized conditions of the simulator. To facilitate the emergence of collision-avoidance behaviors, in addition to modeling dynamics, we simulate collisions between individual quadrotors. In reality, collision outcomes depend on many factors, *e.g.* rigidity and materials of the quadrotor frames, whether the blades of two colliding quadrotors touch, *etc.* Instead of modeling these complex processes, we adopt a simple randomized collision model. When collision between quadrotors is detected, we briefly apply random force and torque to both quadrotors with opposite signs, preserving linear and angular momenta.

## 3.3 Training setup

For training we use a policy gradient RL algorithm. In our setup, the learning algorithm updates the parameters $\theta$ of the policy $\pi_\theta(a^{(t)}|o^{(t)})$ to maximize the expected discounted sum of rewards $\mathbb{E}_{\pi_\theta}\left[\sum_{t'=1}^{T} \gamma^{t'-t} R(S^{(t')}, a^{(t')})\right]$. For our experiments we choose Proximal Policy Optimization (PPO) [28]. In particular, we use the implementation from the high-throughput asynchronous

RL framework "SampleFactory" [29] which supports multi-agent learning and enables large scale experiments. In each training episode, the goal of every quadrotor in the swarm is to reach its designated position in the formation while avoiding collisions with the ground and with other quadrotors. In order to provide rich and diverse training experience, we train our policies in a mixture of scenarios featuring a variety of geometric 2D and 3D formations. Further training details are in Section 4. The reward function optimized by the RL algorithm for quadrotor $i$ consists of three major components: $r^{(t)} = r_{\text{pos}}^{(t)} + r_{\text{col}}^{(t)} + r_{\text{aux}}^{(t)}$. Here, $r_{\text{pos}}^{(t)} = -\alpha_{\text{pos}} \left\| p_i^{(t)} \right\|_2$ rewards the quadrotors for minimizing the distance to their target locations. $r_{\text{col}}^{(t)}$ is responsible for penalizing collisions between quadrotors and is defined as: $r_{\text{col}}^{(t)} = -\alpha_{\text{col}} \mathbb{1}_{\text{col}}^{(t)} - \alpha_{\text{prox}} \sum_{j=1}^{K} \max \left( 1 - \left\| \tilde{p}_{ij}^{(t)} \right\|_2 / d_{\text{prox}}, 0 \right)$. The indicator function $\mathbb{1}_{\text{col}}^{(t)}$ is equal to 1 for timesteps where a new collision involving the $i$-th quadrotor is detected. The second term represents the smooth proximity penalty with linear falloff distance $d_{\text{prox}}$. Here $\left\| \tilde{p}_j^{(t)} \right\|_2$ is the distance between centers of mass of $i$-th and $j$-th quadrotors, and $d_{\text{prox}}$ in our experiments is double the size of the quadrotor frame, which encourages them to keep minimal distance in tight formations. In addition to the first two terms $r_{\text{pos}}^{(t)}$ and $r_{\text{col}}^{(t)}$ that convey our main objective we also adopt an auxiliary reward function similar to [20] to facilitate the initial learning of stabilizing controllers: $r_{\text{aux}}^{(t)} = -\alpha_\omega \left\| \omega_i^{(t)} \right\|_2 - \alpha_f \left\| f_i^{(t)} \right\|_2 + \alpha_{\text{rot}} R_{33}^{(t)}$ penalizing high angular velocity, high motor thrusts, and large rotations about horizontal ($x$- and $y$-) axes respectively.

## 3.4 Model architectures

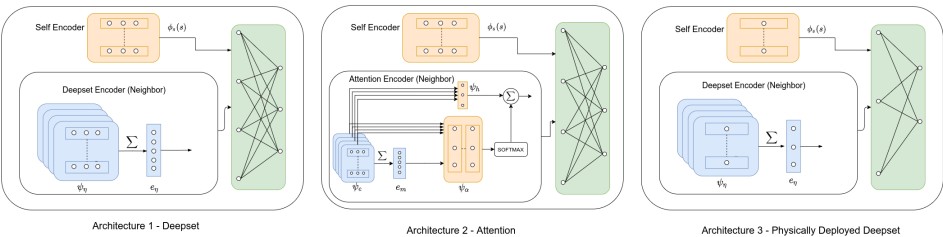

Figure 2: Detailed neural network architectures. Left: Deepsets architecture used in simulation. Middle: Attention architecture used in simulation. Right: Smaller deepsets architecture deployed on Crazyflie2.0.

During training and evaluation, the quadrotors' actions $a$ are sampled from a parametric stochastic policy $\pi_\theta(a|o)$. We omit time indices and quadrotor identity for simplicity where possible. We represent $\pi_\theta$ with a Gaussian distribution, $a \sim \pi_\theta(a|o) \overset{d}{=} \mathcal{N}(\mu_a, \sigma^2)$, where the mean $\mu_a \in \mathbb{R}^4$ is a function of the quadrotor's observation at time $t$ parameterized by a feed-forward neural network, and the variance $\sigma^2 \in \mathbb{R}$ is a single learned parameter independent of the state. To compute the distribution means we embed the state of the quadrotor and its neighborhood before regressing: $\mu_a : e_s = \phi_s(s), \quad e_\eta = f_\eta(s, \eta), \quad \mu_a = \phi_a(e_s, e_\eta)$. Here $e_s$ and $e_\eta$ are the embedding vectors that encode each quadrotor's own state and the state of its neighborhood respectively, $\phi_s$ and $\phi_a$ are fully-connected neural networks, and $f_\eta$ is the *neighborhood encoder*. We analyze two types of neighborhood encoders: deep sets and attention-based (Section 3.5). The value function $V_\pi$ uses the same architecture as the policy, except that it regresses a single deterministic value estimate instead of the parameters of the action distribution. Weights are not shared between $\pi$ and $V_\pi$ models.

## 3.5 Neighborhood encoder

**Deep sets.** The task of the encoder $f_\eta$ is to generate a compact and expressive representation $e_\eta$ of each quadrotor's local neighborhood. Since the individual quadrotors are agnostic to the identity of their neighbors, this representation must be permutation invariant. In addition, scale invariance is a desirable property since the size $K$ of the observable neighborhood fluctuates over time, *i.e.* when a sufficient number of quadrotors in the formation move beyond the sensor range. The deep sets architecture proposed by Zaheer et al. [30] has these required properties. We apply the same learned transformation $\psi_\eta$ to the observed features of each quadrotor in the neighborhood, after which a permutation-invariant aggregation function is applied to neighbor embeddings $e_j$. We calculate the

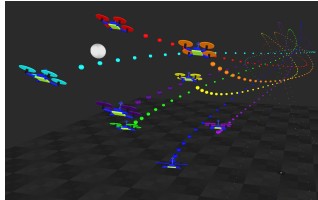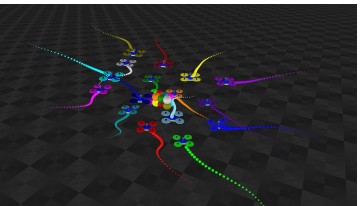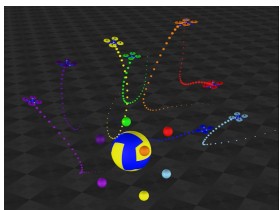

Figure 3: (Left) Evader pursuit, (Middle) $N = 16$ quadrotors in a dense formation after fine-tuning (see Section 4.3), and (Right) A swarm breaking formation to avoid a collision with an obstacle. Videos of the learned swarm behaviors in different scenarios are at: https://sites.google.com/view/swarm-rl.

mean of neighbor embeddings to achieve scale invariance: $e_j = \psi_\eta(\tilde{p}_{ij}, \tilde{v}_{ij})$, $e_\eta = \frac{1}{K}\sum_{j=1}^K e_j$. Not all neighbors are equally important for trajectory planning and decision making. For example, distant and stationary drones are less likely to influence a drone's behavior compared to closely located and fast-moving neighbors. The mean operation in the deep sets encoder does not allow it to convey the relative importance of different neighbors – this motivates a more sophisticated encoder architecture.

**Attention-based.** The attention mechanism [31] provides a natural way to express the relative importance of individual neighbors. The attention-based neighborhood encoder is based on [32], adapted for quadrotors in 3D. The current quadrotor's state $s_i$ and the neighbor observations $(\tilde{p}_{ij}, \tilde{v}_{ij})$ for the $j$-th neighbor are used to compute the attention weights: $e_j = \psi_e(s_i, \tilde{p}_{ij}, \tilde{v}_{ij})$, $e_m = \frac{1}{K}\sum_{j=1}^K e_j$, $\alpha_j = \psi_\alpha(e_j, e_m)$. Here $e_j$ are the embedding vectors of individual neighbors, $e_m$ is the summary of the whole neighborhood, and $\psi_e$ and $\psi_\alpha$ are fully-connected neural networks. We use the softmax operation over $\alpha_j$ to compute the attention scores which sum up to 1. The neighborhood embedding is thus produced as $e_\eta = \sum_{j=1}^K \text{Softmax}(\alpha_j)\psi_h(e_j)$, where $\psi_h$ represents an additional hidden layer. Both in deep sets and attention encoders, we used multi-layer perceptrons (MLPs) with 256 neurons and tanh activations. For additional details see [33].

# 4   Experiments and results

We train our control policies in episodes, in diverse randomized scenarios with static and dynamic formations. Our virtual experimental arena is a $10 \times 10 \times 10$ m room. At the beginning of each episode, we randomly spawn the quadrotors in a 3 m radius around the central axis of the room, at a height between 0.25 and 2 m. We randomly initialize their orientation, linear and angular velocities to facilitate learning robust recovery behaviors. To provide a diverse and challenging training environment, we procedurally generate scenarios of different types, listed below.

**Static formations.** The target formation is fixed throughout the episode. The formation takes various geometric shapes *e.g.* 2D grid, circle, cylinder, and cube (Fig. 1). The separation $r$ between goals in the formation is chosen randomly. A special case $r = 0$ where the goal locations for all quadrotors coincide demands very dense configurations with high probabilities of collisions. We refer to this task as the *same goal* scenario.

**Dynamic formations.** We modify the separation between the quadrotors, and the position of the formation origin. We explore gradually shrinking the inter-quadrotor separation over time, and randomly teleporting the formation around the room. To train policies to avoid head-on collisions at high speed, we include scenarios where the swarm is split into two groups. We swap the target formations of quadrotors in these two groups several times per episode, which requires two teams of quadrotors to fly 'through' each other. We refer to this as the *swarm-vs-swarm* scenario.

**Evader pursuit.** For the evader pursuit task, the team is given a shared goal that moves according to some policy (simulating an evader that the team must pursue). We use two evader trajectory parameterizations: a 3D Lissajous curve and a randomly sampled Bezier curve.

## 4.1   Model architecture study

We compare training performance with different neighborhood encoders with $N = 8$ quadrotors (Fig. 4) and a fixed number $K = 6$ of visible neighbors. In addition to the architectures described

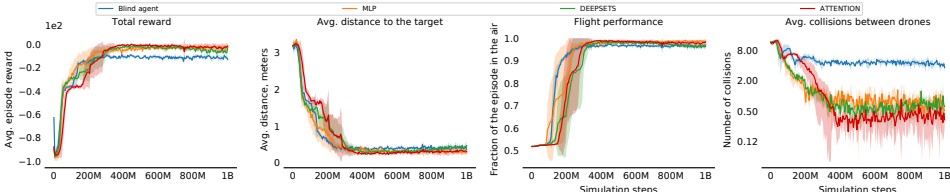

Figure 4: Comparison of different model architectures for $N = 8$ quadrotors with $K = 6$ neighbors visible to each. For each architecture we show mean and standard deviation of four independent training runs.

in Section 3.5 we train two baselines. The first is a *blind* quadrotor, for which we remove the neighborhood encoder $f_\eta$ entirely. While blind quadrotors get close to their targets, they are not able to avoid collisions with each other. The second uses a plain multi layer perceptron, which concatenates neighbor observations as input. This policy, which is not permutation invariant, fails to avoid collisions in most scenarios, suggesting that a permutation-invariant architecture is needed. The difference between the attention and deep sets encoders is most prominent in tasks that require dense swarm configurations, such as the *same goal* scenario. In addition, quadrotors with the deep sets encoder do not get as close to their target locations, sacrificing formation density to minimize collisions. Since the attention-based architecture demonstrated the best performance in both goal reaching and collision avoidance, we use it in all further simulated experiments.

## 4.2 Attention weights study

We investigate the results of training an attention-based architecture for encoding relative scores of neighboring drones. We ask whether the attention mechanism learns to assign higher attention scores to neighbors that are closer and whose velocity vector points towards the current agent. In addition, we investigate to what degree distance and velocity individually affect the scores. We modify the *swarm-vs-swarm* scenario to contain two teams of two drones whose goals are 1 m apart and situated in the same horizontal plane. The drones are allowed to settle at their respective goals following which the goals are swapped. We take a snapshot of the experiment and record the softmax attention weights for each drone. We manually set the relative velocities of all neighbors to $(0, 0, 0)$ for each drone and pass the modified observations to the attention encoder. The results (Fig. 5) show that the red quadrotor assigns the highest attention weight of 0.61 to the blue quadrotor, which is on a collision course with it. Similarly, the blue quadrotor assigns the highest weight of 0.57 to its red neighbor. For the gray and green quadrotors, all neighbors are assigned a roughly equal weighting, with neighbors closer in distance having slightly higher weights. When the relative velocity observations are manually set to 0 and fed to the attention encoder, we observe a drastically different, much more uniform distribution of attention weights. We conclude that neighboring quadrotors with small relative distance and high relative velocity vectors in the direction of the viewer are prioritized over drones further away with velocities in other directions. Drones with relative velocities pointing towards the viewer seem to be prioritized higher than drones that are closer but with velocity vectors away from the viewer, implying that velocity is considered more important than distance.

## 4.3 Scaling

We investigate the ability of our policies to scale to larger swarm sizes without re-training from scratch. We introduce a second, fixed-cost training step in which policies trained with $N = 8, K = 6$

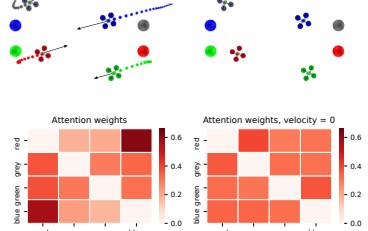

Figure 5: Attention weights between drones. Left: original velocities, right: velocities set to 0.

| # agents | Collisions per minute per drone | Distance to target, m | Collisions per minute per drone +200M training | Distance to target, m +200M training |
|---|---|---|---|---|
| 8 | 0.02 | 0.42 | 0.00 | 0.41 |
| 16 | 0.09 | 0.57 | 0.08 | 0.63 |
| 32 | 0.90 | 0.86 | 0.16 | 0.81 |
| 48 | 2.43 | 1.11 | 0.23 | 0.92 |
| 64 | 5.36 | 1.70 | 0.29 | 1.12 |
| 128 | 8.63 | 4.32 | 1.37 | 3.05 |

Table 1: Scaling up the attention policy with and without additional training. The size of the visible local neighborhood is fixed at $K = 6$ drones. The metrics are averaged over 20 episodes.

are trained for an additional $2 \times 10^8$ steps with the new target number of quadrotors $N$ and the same $K = 6$, *i.e.* the baseline policies are copied and tuned separately in the environment with larger swarm size. The results (Table 1) show that without additional training, the number of collisions increases with the number of quadrotors because the state distribution changes significantly compared to 8-drone case. Additional tuning has a significant positive effect. Even with 128 drones, the quadrotors can avoid collisions in dynamic environments, and the higher number of collisions is largely explained by cascade effects: when a collision happens, it affects multiple drones. The additional tuning amounts to only 20% of the original training session ($\sim$ 4 hours).

## 4.4 Obstacle avoidance

We experiment with a harder version of the environment by introducing a spherical obstacle moving through the formations at random angles multiple times throughout the episode. At the beginning of each episode, we randomly sample the obstacle size, as well as its velocity and the parameters of its trajectory. To incorporate obstacles into our training protocol, we augment the quadrotor observations $o_i^{(t)}$ to contain the information about obstacle state $\zeta_i^{(t)}$. This information includes the radius $\hat{r}$ of the obstacle, and its position $\hat{p}_i^{(t)}$ and velocity $\hat{v}_i^{(t)}$ relative to the $i$-th quadrotor. We process the obstacle observations with an additional MLP $\phi_o$ to produce the obstacle embedding $e_o$, which is used in conjunction with the neighborhood encoder to generate the action distributions (see Section 3.4 for details): $e_o = \phi_o(\zeta), \quad \mu_a = \phi_a(e_s, e_\eta, e_o)$. The collision physics and the penalties are modeled in the same way as for quadrotor-vs-quadrotor collisions (Sections 3.2 and 3.3). Fig. 7 shows the training performance in obstacle avoidance scenarios. Despite increased complexity, we achieve performance comparable to the baseline, keeping dense formations close to the target locations.

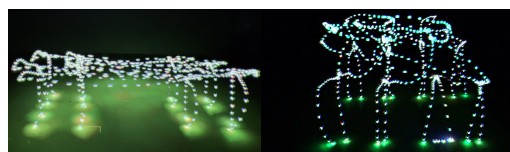

Figure 6: Time lapse of eight drones swapping goals (left) and performing a formation change (right).

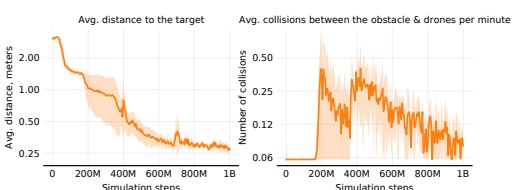

Figure 7: Learning to maintain dense formations while avoiding collisions with moving external obstacles.

## 4.5 Additional baselines

Classical trajectory optimization and control methods have been proposed for tasks similar to ours. [34] uses graph search in discretized space followed by trajectory smoothing to switch formations with large teams of quadrotors. It relies on full prior information about the (static) environment and an offline optimization process taking up to tens of seconds. We test our controller in randomized dynamic environments without access to global information, which makes it hard to make a direct comparison between our method and [34]. Graph Neural Networks [16, 17] and RL [18, 19] are a prime candidate for an alternative model architecture. GNNs can capture global information about the swarm while relying only on local communication by performing multiple consecutive graph convolution operations and communicating intermediate representations between adjacent robots. While this enables decentralized operation, the reliance on multiple message exchanges between UAVs on each step is prohibitively expensive for nano-quadrotor platforms such as Crazyflie. Our controllers run at 500Hz on the real drones, which makes the communication protocol latency requirements exceptionally tight for multi-layer GNNs. Nonetheless, GNNs remain an attractive option for more powerful platforms or tasks that do not require high-frequency reactive control. [10] uses Buffered Voronoi Cells to compute safe regions around quadrotors (with margins between cells to account for kinodynamic constraints) and a PID controller to achieve positions within safe regions that are the closest to the target. This method relies only on local neighborhood information. We implement it on our hardware platform and compare its performance to our method (Section 5).

## 5 Physical deployment

We deployed our policies on the Crazyflie2.0, a small, lightweight, open-source quadrotor platform. We tested the neural controller in several scenarios: hovering in a close proximity to a shared goal

(*same goal*), following a moving target, maintaining dynamic geometric formations, and flying through a team of moving drones (*swarm-vs-swarm*), with up to 8 quadrotors in the latter experiments (Fig. 6). Video demonstrations are available at https://sites.google.com/view/swarm-rl. Crazyflie2.0 is a low-power nano-quadrotor platform where on board computation is provided by a microcontroller with 168MHz and 192Kb of RAM. To run a team-aware neural policy on such limited hardware we trained a much smaller version of our deep sets model with only 16 and 8 neurons in the hidden layers of self- and neighbor encoders respectively. Surprisingly, even such tiny policies with $\sim 10^3$ parameters performed well on real quadrotors. We use a Vicon system to provide position and velocity updates at 100 Hz with an added low-pass exponential filter for neighbor positions to reduce noise. Each drone's controller runs at 500 Hz incorporating the IMU measurements atop the latest available Vicon data. Despite the fact that downwash was not explicitly modeled in the simulator, our policies recover from the aerodynamic disturbances caused by the proximity of other drones. We observed recovery from non-destructive collisions with each other or with the ground wherein drones re-stabilize after collisions with teammates, and "bounce back" from the ground and resume flight.

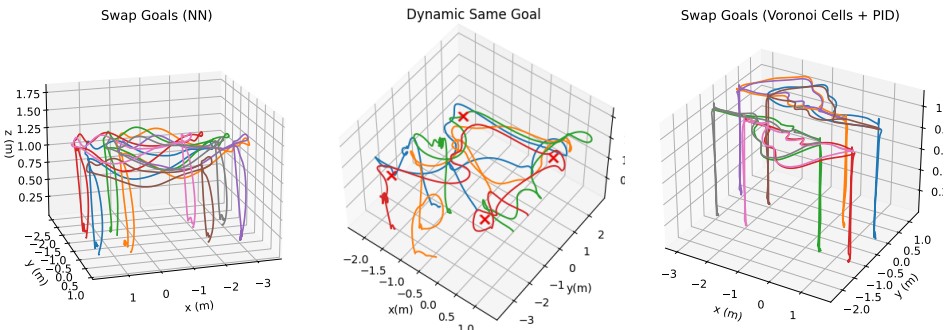

Figure 8: (Left) Two teams of four quadrotors swap goals twice and then land at their original starting positions. (Middle) Four quadrotors follow a moving goal, starting from the bottom right going counter-clockwise. The goal positions are marked with red X's. Both plots show 3D positions over time of the four quadrotors. (Right) two teams of four quadrotors swap goals using PID controllers and buffered Voronoi cells for collision avoidance.

For comparison, we implemented an online planning and collision avoidance algorithm [10], based on buffered Voronoi cells and PID control, to understand how our controllers behave compared to a traditional approach. We tested with 8 drones on the *swarm-vs-swarm* scenario. The trajectories generated by the classical algorithm are longer than the ones produced by our neural network controller. Although there were no restrictions on the configuration space, we observed that the classical method generates trajectories predominately in a single 2D $xy-$plane, whereas the neural controllers utilize all 3 spatial dimensions (Fig. 8). Finally, we observe that our controllers execute more aggressive maneuvers, reaching goals faster with max velocity (acceleration) of up to 4 $m/s$ (7 $m/s^2$) respectively, compared to a max of 1 $m/s$ (3 $m/s^2$) for the classical method. For additional details see [33].

## 6   Conclusion

Our results demonstrate that drones trained with deep reinforcement learning can achieve strong goal-reaching and collision avoidance performance across a diverse range of training scenarios with realistic quadrotor dynamics. We present evidence of successful swarm control in simulation and demonstrate the zero-shot transfer of policies learned in simulation to the Crazyflie platform, on which we are able to perform successful trials on multiple tasks. Our policies learn to fly from scratch, without the use of tuned PID controllers. Our method is thus model-agnostic, *i.e.* we can learn policies for quadrotors with different physical parameters (*e.g.* mass, size, inertia matrix, thrust) by simply re-running the training in the updated simulator. In contrast with classical planning methods, we do not introduce any constraints on the velocity or acceleration and allows the controller to take advantage of full capabilities of the quadrotor. This enables agile flight with aggressive maneuvers. Our pursuit-evasion experiments are the most representative of this. In order to stay close to the fast evader, simulated quadrotors exceed speeds of 7 m/s and reach accelerations up to 1.7 $g$. Permutation and scale-agnostic model architectures used in our method allow us to switch between different team sizes. We found that after additional training to adjust to the larger team size, our policies can control swarms of up to 128 members without a significant increase in computation burden per quadrotor.

**Acknowledgments**

This research is supported in part by a USC Graduate Fellowship. We thank our colleagues from the Robotic Embedded Systems Laboratory (RESL) for discussions and insights. We are grateful to Baskın Şenbaşlar, Tao Chen, and Wolfgang Hönig for their assistance and support. We particularly thank James Preiss for his help with the experimental setup.

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
