# OpenReview forum: "Decentralized Control of Quadrotor Swarms with End-to-end Deep Reinforcement Learning"
_robot-learning.org/CoRL/2021/Conference — CoRL2021 Poster_

### Official Review · Reviewer_YbEi · 2021-07-18

**Originality:** Good
**Technical Quality:** Good
**Clarity Of Presentation:** Very Good
**Impact:** 4

**Recommendation:**

Weak Accept: I recommend accepting the paper, but will not argue for my recommendation if the majority of other reviewers have a different opinion.

**Summary:**

Main Contributions
===============
- Successful distributed swarm control for drones in simulation and the real world.
- Analysis of different network architectures on final performance.

Summary
============
The paper presents successful decentralized drone swarm control using a learned policy. A group of drones is tasked with maintaining static or dynamic formation flight while avoiding collisions with each other or external, possibly dynamic, obstacles. Each drone is controlled by a policy represented by a deep neural network, that observes the current state of the drone, the state of neighboring drones as well as the state of potential obstacles.
The control policy is trained using deep reinforcement learning (PPO).
The paper compares different model architectures with respect to the final performance of the drone swarm.
The proposed approach is successfully deployed in simulation and in the real world.

**Issues:**

The paper would get significantly stronger if the authors included a comparison against prior work (see above for potential baselines).

**Reviewer Expertise:**

Good: General knowledge of the area

**Strengths And Weaknesses:**

Strengths
===========
- The paper demonstrates successful deployment of the proposed approach to a real-world setting with 4 crazyflie quadrotors.
- The paper presents an in-depth analysis of different network architectures.
- The results are very well illustrated in the supplementary video.
- The paper is well written and easy to follow.

Weaknesses
==========
- Even though the presented results are impressive, the paper does not compare against prior work. All experiments are performed by comparing the proposed approach against different variants of itself (i.e. a 'blind' quadrotor, that has no access to the position of neighboring vehicles).
- To better understand the performance of the proposed approach, the paper would significantly gain value if it compared with prior work on drone swarm control such as [1], or even naive baselines based on reactive navigation [2] or potential fields.

[1] Hönig, W., Preiss, J.A., Kumar, T.S., Sukhatme, G.S. and Ayanian, N., 2018. Trajectory planning for quadrotor swarms. IEEE Transactions on Robotics, 34(4), pp.856-869.

[2] Florence, P., Carter, J. and Tedrake, R., 2020. Integrated perception and control at high speed: Evaluating collision avoidance maneuvers without maps. Algorithmic Foundations of Robotics XII. Springer, pp.304-319.

**Summary Of Recommendation:**

The paper shows convincing results, but lacks any comparison to even naive baselines.

---

> ### Author Response · Authors · 2021-08-31
> **Response to Reviewer YbEi**
>
> Thank you for your review! We acknowledge that baselines would help better place our work in the existing literature and thus ran a limited version of Buffered Voronoi Cell algorithm. These results have been included in the revised manuscript, specifically sections 4.5 and Supplementary Materials section IV. Regarding [1], this paper proposes a centralized, offline planner to compute downwash-aware collision-free trajectories for quadrotor swarms in static environments. Our approach, however, is completely decentralized, allowing for online re-planning and control. In our problem domain, we introduce scenarios such as evader-pursuit and same static goal where collisions are highly likely and dynamic replanning is required, with the intent to learn robust policies capable of aggressive maneuvers and cooperative collision avoidance behaviors. [1] does not allow for dynamic replanning and is thus infeasible for several of our scenarios such as evader-pursuit and dynamic obstacle avoidance. We believe that our approach and [1] provide solutions to fundamentally different problems and thus decided that [1] would not be a suitable baseline.
>
> [2] Integrates perception and control to enable fast, reactive control through obstacle-dense environments. While this approach is similar in that it enables dynamic replanning and collision avoidance, it utilizes depth information from a camera, which is not implemented in our simulator because it is not present on our crazyflie platform. In addition, [2] outlines an approach to high speed, reactive control for a single quadrotor, whereas we are concerned with emergent cooperative behaviors and aggressive maneuvers in the multi-agent setting. Thus, we do not believe this to be a suitable baseline as well.

---

> > ### Comment · Reviewer_YbEi · 2021-09-03
> > **Post-Rebuttal Response**
> >
> > Thank you for the revised version.
> > After taking the changes in the manuscript and the authors' response into account, I keep my assessment of 'weak accept'. While the authors provide a reason to not compare against simple traditional baselines, I see this as a missed opportunity to showcase the performance of the proposed approach. Especially for [2], the core idea of this algorithm is not that it uses depth information, but that it leverages a simple library of motion primitives. It would be interesting to see how such a simplistic approach (each quadrotor individually plans its actions using a library of primitives while assuming other agents to be static obstacles) would perform on the tasks presented in this work.

---

### Official Review · Reviewer_g2nt · 2021-07-19

**Originality:** Good
**Technical Quality:** Good
**Clarity Of Presentation:** Very Good
**Impact:** 3

**Recommendation:**

Weak Accept: I recommend accepting the paper, but will not argue for my recommendation if the majority of other reviewers have a different opinion.

**Summary:**

This paper proposes a multi-agent end-to-end reinforcement learning approach to learning controllers for drone swarms that achieve formation control goals and collision avoidance using only local (and hence scalable) policies.  The policies are trained in a realistic simulated environment, and then deployed in both simulation and on Crazyflie 2.0 physical testbeds.  Each drone's policy learns and acts o a local embedding of the drone's state, an embedding of its neighbors' relative position and velocity -- for the neighbor encoder, Deepset and attention-based architectures are compared, and it is shown that the latter achieves better performance by by learning to suitably weight "more dangerous" neighbors (i.e., those that are closer and heading towards the ego-drone) more than "less dangerous" ones.  The policies are trained using PPO on several procedurally generated scenarios, including static and dynamic formations, as well as evader pursuit like settings, and the proposed method is also extended to accommodate obstacle avoidance.  Experiments evaluating the two neighbor embedding architectures, the attention weights, and scalability of the approach are conducted.  Finally, scaled-down versions of the Deepset architecture are trained in simulation and then deployed and evaluated on a computationally constrained physical platform (four Crazyflie 2.0 drones).

**Issues:**

Comparison to stronger baselines, and clarification of scalability of training time.  See comments above for more details.

**Reviewer Expertise:**

Good: General knowledge of the area

**Strengths And Weaknesses:**

Strengths:
+ the paper is well written and easy to follow
+ the first successful demonstration of multi-agent end-to-end deep RL for drone swarm control is indeed novel
+ the use of an attention mechanism for distinguishing "important" drones from "unimportant" ones given a specific dynamic configuration is a natural and nice idea
+ the experiments covered a broad range of interesting tasks and demonstrated that the method learned successful collision free formation control policies.

Weaknesses:
- the baselines against which the method in the paper is compared seem to be quite weak.  The "blind" baseline (where the quadrotors can't see their neighbors) is too weak to be meaningful, and whereas the MLP (non-permutation-invariant) policy is sensible, what's missing are some more sophisticated baselines for comparison.  For example, in the related work section, the paper explains that [9,10] significantly constrain the configuration space of the robot to guarantee collision free behavior.  It would have been nice to see a comparison of collision frequency vs. goal tracking performance for the methods of [9,10] and those proposed in the paper.  Similarly, for permutation-invariant policies, recent lines of work have shown that Graph Neural Networks (GNNs) can be used to achieve scalable learning and control in the context of formation control.  See for example:

@inproceedings{tolstaya2020learning,
  title={Learning decentralized controllers for robot swarms with graph neural networks},
  author={Tolstaya, Ekaterina and Gama, Fernando and Paulos, James and Pappas, George and Kumar, Vijay and Ribeiro, Alejandro},
  booktitle={Conference on robot learning},
  pages={671--682},
  year={2020},
  organization={PMLR}
}

and

@inproceedings{khan2020graph,
  title={Graph policy gradients for large scale robot control},
  author={Khan, Arbaaz and Tolstaya, Ekaterina and Ribeiro, Alejandro and Kumar, Vijay},
  booktitle={Conference on robot learning},
  pages={823--834},
  year={2020},
  organization={PMLR}
}

(Disclaimer: I am not an author on either paper).

- the argument for scalability didn't strike me as convincing: if 500 million simulation steps are needed to refine policies trained with a smaller number of drones to a larger number of drones, this seems like it will run into computational bottlenecks if the number of drones starts getting large.  I may have misunderstood this point, so would like to see some clarification in the rebuttal: for example, showing wall-clock time for the fine-tuning step as a function of the number of drones would be helpful.


**Summary Of Recommendation:**

Suggesting acceptance based on: first demonstration of end-to-end multi-agent RL to collision free formation control, well written paper, interesting experiments (including sim2real validation).

Not suggesting strong accept based on: weak baselines and unconvincing argument of training scalability.

---

> ### Author Response · Authors · 2021-08-31
> **Response to Reviewer g2nt**
>
> Thank you for your comments! We have updated the manuscript to include [10] as a baseline. Regarding [9], this paper largely ignores kinodynamic and dynamic constraints of different robot platforms. In our approach, we aim to learn policies capable of aggressive maneuvers and tight formations with minimal collisions and thus utilize a complex dynamics model in our simulation. In addition, our simulation is 3D and introduces dynamic obstacles, which are not mentioned in the paper. As such, we concluded it would be challenging to provide a direct comparison with this method without major modifications.
>
> Regarding permutation-invariant learned policies, as you pointed out, GNNs excel at exploiting local structure in the graph and incorporating information from robots multiple hops away. However, our intent from the start was to employ algorithms and construct a simulator specifically for transferring policies onto computationally constrained hardware. Our controllers run at high frequencies and only require local sensing of the drone’s neighborhood to run a highly condensed MLP that outputs direct thrust commands. Since GNNs require communicating information with the local neighborhood, it becomes very difficult to maintain such a high control frequency with such limited compute. We think that GNNs would be better suited for a heavier platform with greater compute and communication capabilities.
>
> Regarding scalability, we ran additional experiments in which the policy for 8 drones was trained for an additional 200 million steps with up to 128 drones, the results of which have been updated in Fig. 6. This additional training step represents only 20% of the original policy’s training time (about 4-5 hours of wall time depending on the swarm size) and still achieves relatively low collisions per minute per drone. In addition, for all our scaling experiments up until 128 drones, we only use observations from the K=6 nearest neighbors and not the N total number of drones in the swarm. Thus, the computational complexity of our approach is a function of the fixed value K and not N. We believe these two components, namely only a small fixed additional training step and a fixed number of local observations for each drone, merit scalability.

---

### Official Review · Reviewer_s99C · 2021-07-29

**Originality:** Poor
**Technical Quality:** Fair
**Clarity Of Presentation:** Good
**Impact:** 3

**Recommendation:**

Strong Reject: I recommend rejecting the paper and will argue for my recommendation even if other reviewers hold a different opinion.

**Summary:**

This paper proposes using RL for decentralized quadrotor control.

**Issues:**

I would recommend the authors spend time looking into the works stated above and think more carefully about where their methods/paper would fit in.


**Reviewer Expertise:**

Excellent: Expert knowledge on the topic of the paper

**Strengths And Weaknesses:**

This paper poses using deep RL for decentralized control. While the paper is well written, it is missing quite a few advances in the field
of large-scale decentralized robot control using RL/Imitiation learning (see citations below). I would like to expand this review on the basis of the following; novelty, quality and clarity

Novelty: This paper proposes using a deep RL method for learning multi-agent RL policies. This is by no means new and there are many many papers that have expanded this field. The first of these works look to use off-the-shelf multi-agent RL policies by parametrizing each agent using a MLP and then using a centralized critic architecture [1]. These papers realize that these works having issues when attempting to scale to a larger number of robots due to the increase in dimensionality during training. The architectures proposed in this paper suffer from the same issues; as the number of robots is increased the number of inputs to the final "green" blocks in Figure 2 increases thus increasing the dimenisonality and as such making these methods not scalable. Further, the methods proposed in [2]-[4] and the papers citing them use graph neural networks to introduce sparsity into the problem while exploiting local structure. These papers produce state-of-the-art results for a large number of robots upto 100). The fact that this paper neither cites these papers nor compares against these methods reflects poorly. As such in my opinion the paper introduces no novelty and no significant contribution compared to the existing state-of-the-art methods for large scale decentralized robot control.

Quality: The paper is well written. However, as mentioned above the experimental section in this paper is underwhelming. The authors show no comparisons with other "non-author proposed architectures", i.e all baselines in Fig 4 consist of some modification proposed by the authors and as such there is no idea how these would compare with other state-of-the-art methods. Fig 6 states "baseline"; I am unsure which baseline this is. Similar issue with Fig 7. There are no meaningful baselines and as such, I fail to see the merits of this approach.


[1] Khan, Arbaaz, et al. "Learning safe unlabeled multi-robot planning with motion constraints." 2019 IEEE/RSJ International Conference on Intelligent Robots and Systems (IROS). IEEE, 2019.
[2] Graph Neural Networks for Decentralized Multi-Robot Path Planning Q.Li et. al
[3] Learning Decentralized Controllers for Robot Swarms with Graph Neural Networks Tolstaya et. al
[4] Graph Policy Gradients for Large Scale Robot Control , Large Scale Distributed Collaborative Unlabeled Motion Planning With Graph Policy Gradients Khan et. al



**Summary Of Recommendation:**

As such, since the paper does not improve/cite upon the existing state-of-the-art methods, for me it adds very little to the existing body of work that looks to use machine learning for teams of robots. As such I cannot recommend this paper be accepted in its current form.

---

> ### Author Response · Authors · 2021-08-31
> **Response to Reviewer s99C**
>
> Thank you for your review and your comments! We have updated our manuscript with citations to the papers that you mentioned and we apologize for not representing this body of work in the original version.
>
> We acknowledge that our proposed MARL methods are not novel, in fact we deliberately chose well-established learning techniques and permutation-invariant architectures that we deemed suitable for our domain. While the architectures and algorithms are indeed not novel, we believe that our application and systems contribution is novel: we demonstrated feasibility of end-to-end RL for decentralized swarm control and agile flight that is also zero-shot transferable to a computationally-constrained real world platform. We made several changes in the manuscript to highlight this point.
>
> Regarding scalability: indeed the chosen MARL architectures would scale poorly if we utilized all information about the swarm. To address this, we use “local observations”, i.e. each drone observes K closest drones, where K < N (total number of drones). Together with the fact that we don’t use a centralized critic, computational complexity of both inference and training becomes independent of N. We mention local observations in sections 3.1 and 4.1 of both original and revised manuscript, and we added additional references to this in the revised manuscript, i.e in the section on scaling. Even with K << N (i.e. K=6 N=128) our controllers are still capable of agile flight with infrequent collisions (Figure 6). Overall, our method can scale to large swarms. With a more careful choice of formations it is possible to scale further, although simulation can become a bit slower (the computational complexity of training and inference remains the same).
>
> To address questions about comparisons with GCNs [3][4]: first of all, thank you for bringing this to our attention, we updated the manuscript to properly represent this highly relevant work, specifically in sections 2 and 4.5. GCNs are very attractive because they can incorporate information from a large graph neighborhood while relying only on local connectivity. As a downside, this requires message passing between adjacent nodes in the graph (drones, in our case) to aggregate information beyond one hop. Since our main focus was to develop a method that allows transfer to a limited hardware platform, we focused on methods that do not rely on communication between quadrotors. Our controllers run at high frequency (100Hz or more), and rely only on the information from the immediate neighborhood, because in Crazyswarm it would be very hard to achieve such high-frequency message passing required by multi-layer GCNs between non-trivial number of drone pairs.
>
> In a way, our architectures can be considered a very special case of GCN approach where the graph structure during each timestep is fixed (each drone is connected to exactly K closest neighbors), and only a single neighborhood aggregation operation is performed (one hop). That said, we recognize that the GCN approach is more general, and would be better suited for heavier platforms with better compute and communication capabilities.

---

### Comment · Area_Chair_5qhw · 2021-08-17
**meta-review - opportunities for improvement follow-up**

I wanted to provide a clarification of my comments below, in particular, where I had said “However, unfortunately, that accomplishment alone does not merit publication.”. Getting deep NN’s to work on real robots is hard, and systems papers are very much worthwhile of publication. It could also be the case that I have failed to fully grasp all the challenges involved from the current version of the paper. If this is the case, as an alternate direction, it would be helpful for the revision to focus more on detailing the challenges involved in the deployment and surveying previous attempts to do so in literature.

---

### Meta-Review · Area_Chair_5qhw · 2021-08-14

**Recommendation:** Accept (Poster)
**Confidence:** 5

**Metareview:**

Summary: This paper proposes using a deep RL algorithm for decentralized formation control of aerial robots. The algorithm is able to accommodate static and dynamic formations in the presence of state or dynamic obstacles. The method is validated with simulation and real-world experiments on four off-the-shelf consumer drones.

Clarity: The paper is exceptionally well-written and presents a clear, logical narrative through the motivation, design, and experimental validation of the proposed method.

Quality: Reviewers noted that deploying a deep RL algorithm on board real quadrotors is a notable achievement and a clear contribution of the paper. However, reviewers also noted that the work in its current form leaves open questions about scalability and comparison to more appropriate baselines.

Significance/Originality: Feedback on the contribution of the proposed method is slightly mixed. In particular, it seemed like reviewers were least convinced about how the proposed method stacked up against the existing state-of-the-art.

Pros:
Reviewers seemed aligned that deploying a deep learning algorithm on board real robots is rather impressive. The paper also clearly communicates the proposed approach, but in the text and supplementary video.

Main opportunities for improvement:
In its current form, the revision should address three major themes in the reviewers’ comments:

- More comprehensive literature review: Broadly speaking, this paper did not convince reviewers that it has a place in the existing literature, at least partly because all three reviewers noted specific missing citations.

- All three reviewers commented that the baselines used in the experimental results section are inappropriate. Namely, they compare the proposed algorithm against variants of itself rather than other state-of-the-art algorithms. The revision should compare against more meaningful baselines.

- A major consideration in swarm coordination algorithms, especially decentralized ones, is scalability. Reviewers s99C and g2nt both expressed serious concerns about the method’s ability to scale to the order of hundreds of robots. The revision should either include stronger evidence of the method’s scalability or characterize and make a case for why the team size the algorithm does scale to is still a meaningful “swarm”.

This paper should be seriously commended for being the first to show learned policies on physical quadrotors. However, unfortunately, that accomplishment alone does not merit publication. The rebuttal should still address three key points highlighted above and in reviewers’ feedback - a more complete literature review, comparison against more rigorous baselines, and justification of the technique’s scalability (and exactly how scalability should be defined in this case).

Thank you for considering our feedback, and we look forward to seeing the updated paper.


====== Final Decision

I am pleased to accept this paper as part of CoRL 2021. I would like to sincerely thank the authors for the time they put into revising the paper. The authors thoughtfully addressed reviewer feedback and the "opportunities for improvement" listed in the original meta-review. Particularly impactful were the additional experiments. The simulation results for teams of 64 and 128, in my opinion, definitively addresses scalability concerns. Experiments against the buffered Voronoi cells baseline also added useful insight. I am also excited to see DNNs running on real robots; there is significant "invisible" debugging and design work to get that to work.

---

### Decision · Program_Chairs · 2021-09-13

**Decision:**

Accept (Poster)

**Comment:**

Summary: This paper proposes using a deep RL algorithm for decentralized formation control of aerial robots. The algorithm is able to accommodate static and dynamic formations in the presence of state or dynamic obstacles. The method is validated with simulation and real-world experiments on four off-the-shelf consumer drones.

Clarity: The paper is exceptionally well-written and presents a clear, logical narrative through the motivation, design, and experimental validation of the proposed method.

Quality: Reviewers noted that deploying a deep RL algorithm on board real quadrotors is a notable achievement and a clear contribution of the paper. However, reviewers also noted that the work in its current form leaves open questions about scalability and comparison to more appropriate baselines.

Significance/Originality: Feedback on the contribution of the proposed method is slightly mixed. In particular, it seemed like reviewers were least convinced about how the proposed method stacked up against the existing state-of-the-art.

Pros:
Reviewers seemed aligned that deploying a deep learning algorithm on board real robots is rather impressive. The paper also clearly communicates the proposed approach, but in the text and supplementary video.

Main opportunities for improvement:
In its current form, the revision should address three major themes in the reviewers’ comments:

- More comprehensive literature review: Broadly speaking, this paper did not convince reviewers that it has a place in the existing literature, at least partly because all three reviewers noted specific missing citations.

- All three reviewers commented that the baselines used in the experimental results section are inappropriate. Namely, they compare the proposed algorithm against variants of itself rather than other state-of-the-art algorithms. The revision should compare against more meaningful baselines.

- A major consideration in swarm coordination algorithms, especially decentralized ones, is scalability. Reviewers s99C and g2nt both expressed serious concerns about the method’s ability to scale to the order of hundreds of robots. The revision should either include stronger evidence of the method’s scalability or characterize and make a case for why the team size the algorithm does scale to is still a meaningful “swarm”.

This paper should be seriously commended for being the first to show learned policies on physical quadrotors. However, unfortunately, that accomplishment alone does not merit publication. The rebuttal should still address three key points highlighted above and in reviewers’ feedback - a more complete literature review, comparison against more rigorous baselines, and justification of the technique’s scalability (and exactly how scalability should be defined in this case).

Thank you for considering our feedback, and we look forward to seeing the updated paper.


====== Final Decision

I am pleased to accept this paper as part of CoRL 2021. I would like to sincerely thank the authors for the time they put into revising the paper. The authors thoughtfully addressed reviewer feedback and the "opportunities for improvement" listed in the original meta-review. Particularly impactful were the additional experiments. The simulation results for teams of 64 and 128, in my opinion, definitively addresses scalability concerns. Experiments against the buffered Voronoi cells baseline also added useful insight. I am also excited to see DNNs running on real robots; there is significant "invisible" debugging and design work to get that to work.